# Interferon-β Activity Is Affected by S100B Protein

**DOI:** 10.3390/ijms23041997

**Published:** 2022-02-11

**Authors:** Alexey S. Kazakov, Alexander D. Sofin, Nadezhda V. Avkhacheva, Evgenia I. Deryusheva, Victoria A. Rastrygina, Maria E. Permyakova, Vladimir N. Uversky, Eugene A. Permyakov, Sergei E. Permyakov

**Affiliations:** 1Institute for Biological Instrumentation, Pushchino Scientific Center for Biological Research of the Russian Academy of Sciences, Institutskaya Str., 7, Pushchino, 142290 Moscow, Russia; fenixfly@yandex.ru (A.S.K.); alsofin@mail.ru (A.D.S.); avkhacheva@gmail.com (N.V.A.); janed1986@ya.ru (E.I.D.); certusfides@gmail.com (V.A.R.); mperm1977@gmail.com (M.E.P.); epermyak@yandex.ru (E.A.P.); 2Department of Molecular Medicine, USF Health Byrd Alzheimer’s Research Institute, Morsani College of Medicine, University of South Florida, Tampa, FL 33612, USA

**Keywords:** cytokine, interferon, S100B, protein–protein interaction, cancer, neurological diseases

## Abstract

Interferon-β (IFN-β) is a pleiotropic cytokine secreted in response to various pathological conditions and is clinically used for therapy of multiple sclerosis. Its application for treatment of cancer, infections and pulmonary diseases is limited by incomplete understanding of regulatory mechanisms of its functioning. Recently, we reported that IFN-β activity is affected by interactions with S100A1, S100A4, S100A6, and S100P proteins, which are members of the S100 protein family of multifunctional Ca^2+^-binding proteins possessing cytokine-like activities (Int J Mol Sci. 2020;21(24):9473). Here we show that IFN-β interacts with one more representative of the S100 protein family, the S100B protein, involved in numerous oncological and neurological diseases. The use of chemical crosslinking, intrinsic fluorescence, and surface plasmon resonance spectroscopy revealed IFN-β binding to Ca^2+^-loaded dimeric and monomeric forms of the S100B protein. Calcium depletion blocks the S100B–IFN-β interaction. S100B monomerization increases its affinity to IFN-β by 2.7 orders of magnitude (equilibrium dissociation constant of the complex reaches 47 pM). Crystal violet assay demonstrated that combined application of IFN-β and S100B (5–25 nM) eliminates their inhibitory effects on MCF-7 cell viability. Bioinformatics analysis showed that the direct modulation of IFN-β activity by the S100B protein described here could be relevant to progression of multiple oncological and neurological diseases.

## 1. Introduction

Interferon-β (IFN-β) is a four-helical cytokine secreted in response to various pathogens [1,2,3]. It possesses immunomodulatory, anti/pro-inflammatory, antiviral, anti/pro-microbial, and antitumor activities [1,4,5,6,7]. IFN-β and its derivatives are commonly used in treatment of multiple sclerosis (MS) [8]. Clinical studies confirmed the efficacy of IFN-β use in therapy of cancer, SARS-CoV-2, and asthma [9,10,11]. However, pro-survival products of interferon-stimulated genes can promote suppression of the antitumor effects of IFN-β [12]. Therapeutic use of IFN-β can be accompanied by the onset of fever, chills, fatigue, headache, liver dysfunction, depression, and cutaneous adverse reactions [13,14,15,16]. Despite direct antimicrobial activity [17], IFN-β favors survival of some pathogenic species in host cells [18,19]. IFN-β activated microglia and initiated neuroinflammation and synapse loss in a murine model of Alzheimer’s disease [20]. Furthermore, IFN-β was shown to contribute to psoriasis [21] and sepsis [22] progression. A preclinical study showed that inhibition of IFN-β signaling in traumatic brain injury reduces neuroinflammation, attenuates neurobehavioral deficits, and limits the tissue loss [23]. The disease- and stage-dependent IFN-β activity necessitates elaboration of the therapeutic modalities aimed at compensation of the negative effects of IFN-β. Apparently, we need deeper insight into regulatory mechanisms of IFN-β functioning for that.

IFN-β signals through its cell surface receptors, the IFNAR1-IFNAR2 complex or IFNAR1 alone, thereby inducing expression of numerous interferon-stimulated genes via JAK-STAT or alternative signaling pathways [24,25,26]. Despite the abundance of information on the downstream signaling events and the factors affecting IFN-β expression [2,3,27], the information on the direct modulation of IFN-β activity by extracellular soluble factors remains scarce. To our knowledge, such cases are limited to IFN-β interactions with some representatives of the S100 protein family of multifunctional Ca^2+^-binding proteins with cytokine-like activities [28,29], including S100P [30] and S100A1/A4/A6 proteins [31]. The in vitro data evidence high-affinity IFN-β binding to the Ca^2+^-bound monomeric S100 proteins with equilibrium dissociation constants, K_d_, of 0.1–1 nM [30,31], which is close to the K_d_ value 0.1 nM for the complex of IFN-β with the IFNAR2 extracellular domain [32]. The binding of S100A1/A4/P to IFN-β inhibits suppression of MCF-7 cell viability by IFN-β [30,31]. In the present work, we show that one more member of the S100 protein family, the S100B protein, binds to IFN-β with even higher affinity, exerting more pronounced effect on the activity of this cytokine.

S100B is a 10.5 kDa Ca^2+^-binding protein, expressed in brain, skin, breast, placenta, adipose tissue, kidney, colon, and testis [33]. S100B comprises a low-affinity pseudo EF-hand (K_d_ of 0.2–0.5 mM) and a canonical EF-hand (K_d_ of 10–60 µM), connected by a ‘hinge’ region (reviewed in ref. [34,35,36]). S100B forms a non-covalent dimer, in which terminal helices associate in an antiparallel manner. Zn^2+^ binding increases S100B affinity to Ca^2+^ by an order of magnitude [34]. Ca^2+^ binding to S100B induces solvent exposure of the hydrophobic residues involved in target binding [35]. Meanwhile, some S100B interactions are Ca^2+^-independent [37]. Localization of S100B in cytosol/nucleus and its secretion favors S100B interaction with a broad spectrum of targets, including over 50 proteins [34,38]. This is illustrated by Figure 1, representing the S100B-centered PPI network generated by STRING. This extended PPI network of S100B protein enables regulation of numerous vital processes, such as cell proliferation, differentiation and locomotion, cytoskeleton organization, Ca^2+^ homeostasis, protein degradation and phosphorylation, receptors, enzymes and transcription factors [36]. S100B is associated with multiple sclerosis, cancer, SARS-CoV-2, acute neural injury, Alzheimer’s and Parkinson’s diseases, amyotrophic lateral sclerosis, schizophrenia, epilepsy, bipolar disorder, depression, obesity, diabetes, inflammatory bowel disease, psoriasis, muscular dystrophy, and uveal and retinal disorders [39,40,41,42,43,44]. S100B is clinically used for diagnostics of melanoma, breast cancer, and brain injury [45]. Blocking of S100B binding to p53, RAGE, and other proteins is considered as a promising approach to cancer therapy [46].

## 2. Results and Discussion

### 2.1. High-Affinity Interaction between IFN-β and S100B

To qualitatively probe the S100B interaction with IFN-β (5 μM), their mixtures (S100B to IFN-β molar ratios of 2:0, 0:1, 0.5:1, 1:1, 2:1) were treated with EDAC/sulfo-NHS at 25 °C for 1.5 h, and the resulting cross-linked protein species were analyzed by SDS-PAGE (Figure 2). The cross-linked IFN-β sample reveals the presence of monomeric and oligomeric forms, while the cross-linked apo-S100B (10 μM) is mostly dimeric with noticeable contribution of the monomer. The mixture of IFN-β and apo-S100B (molar ratio of 1:2) exhibits a slight accumulation of S100B monomer, and considerable ‘shift’ and bleaching of the band corresponding to the dimer and accumulation of the band(s) nearby 100 kDa, while the band of monomeric IFN-β remains intact. Nevertheless, the absence of additivity of the bands corresponding to IFN-β and apo-S100B evidence formation of their complex. Meanwhile, Ca^2+^-bound S100B demonstrates somewhat different behavior. Similarly, the cross-linked Ca^2+^-loaded S100B (10 μM) reveals the bands corresponding to its monomer and dimer. The addition of 5 μM IFN-β induces marked bleaching of the bands of S100B and IFN-β monomers, and ‘shift’ and bleaching of the band corresponding to the S100B dimer and accumulation of the band(s) nearby 100 kDa. These effects evidently reflect formation of the S100B–IFN-β complex. The decrease in the S100B level expectedly causes gradual recovery of the band corresponding to monomeric IFN-β. The signs of S100B–IFN-β interaction are noticeable even at S100B to IFN-β molar ratio of 0.5:1. Overall, the crosslinking experiments indicate an S100B interaction with IFN-β with an equilibrium dissociation constant reaching at least the micromolar level. The differences between the results of the experiments performed in the absence/presence of Ca^2+^ indicate a dependence of this interaction on calcium level.

In order to quantitate the interaction between S100B and IFN-β (0.5 μM), the reaction was monitored by the tryptophan fluorescence of IFN-β (three Trp residues; excitation at 295 nm) in the course of its titration by S100B (lacks Trp residues) (Figure 3). The titration of IFN-β by Ca^2+^-bound S100B is accompanied by a minor (*ca* 3 nm) blue shift of the fluorescence emission maximum (Figure 3A), which seems to reflect a slight decrease in the solvent accessibility of the emitting Trp residue(s) of IFN-β [48]. Nevertheless, the final position of the spectrum maximum of 340 nm shows that side chains of the emitting tryptophans in this state of the protein remain highly accessible to the solvent. The S100B binding to IFN-β is accompanied by the quenching of tryptophan fluorescence of IFN-β with the characteristic bend at an S100B to IFN-β molar ratio of 1:1 (Figure 3B). This means that the Ca^2+^-loaded S100B dimer [49] binds two IFN-β molecules. The fluorescence titration data were fitted to the one-site binding model (Figure 3B) with an apparent equilibrium dissociation constant of 21 ± 3 nM, in accord with the chemical crosslinking estimate. Calcium removal from the solution by addition of EDTA results in the disappearance of the S100B-induced effects (Figure 3), which indicates that apo-S100B does not interact with IFN-β. This conclusion contradicts the chemical crosslinking data (Figure 2). The contradiction can be resolved considering that the crosslinking samples could be contaminated by Ca^2+^ (as in the related experiments, calcium chelators were avoided, since their carboxylic groups interact with EDAC/sulfo-NHS). The intriguing observation is the 3 nm difference between fluorescence emission maximum values for IFN-β in excess of Ca^2+^/EDTA (Figure 3A), which could arise due to IFN-β interaction with Ca^2+^ or EDTA. Examination of the crystal structure of human IFN-β (PDB entry 1AU1) reveals the presence of patches of the negatively charged residues, which could comprise a non-canonical Ca^2+^-binding site (for example, residues E53, D54, and E103, E104, E107, E109, D110).

To characterize the S100B–IFN-β interaction by an alternative technique, S100B was immobilized on the surface of an SPR sensor chip by amine coupling and a set of injections of IFN-β solutions (10–80 nM) was conducted. The SPR sensograms for Ca^2+^-bound S100B exhibited a concentration-dependent pattern (Figure 4). The surface of the SPR chip was regenerated by passage of 5 mM EDTA, indicating that apo-S100B lacks affinity to IFN-β, in accord with the fluorometric data (Figure 3). The dissociation phase of the sensograms for Ca^2+^-loaded S100B (Figure 4) reveals two distinct dissociation processes: (1) a relatively fast process with half-life time (t_1/2_) of 100 s; (2) a much slower process with t_1/2_ value exceeding 1000 s. The kinetic SPR data are well approximated by the heterogeneous ligand model [2] (Figure 4) with equilibrium dissociation constants, K_d_, of 47 pM and 87 nM (Table 1). For comparison, the lowest K_d_ values reported for S100B complexes with fragments of its canonical extracellular target, RAGE, are in a range of 2 nM [50].

The drastically lower SPR estimate of the K_d_ compared to that derived from the fluorometric titration (Figure 3) likely arises due to conversion of the S100B dimer [49] into the monomeric state. Similarly, conversion of S100A1/A4/A6/P proteins into monomeric forms increased their affinity to IFN-β by at least two orders of magnitude [30,31]. It is noteworthy that the use of the opposite experiment setting with IFN-β immobilization on the surface of the SPR chip did not reveal S100B–IFN-β interaction at a Ca^2+^-loaded S100B concentration of 5 µM [31], which could be related to the insufficient accuracy of the SPR instrument. Overall, IFN-β binds S100B depending on Ca^2+^ and the oligomeric state of S100B, with a preference to the monomeric Ca^2+^-bound form. The strong calcium dependence of this interaction is likely due to the Ca^2+^-induced solvent exposure of hydrophobic surfaces of S100B, typical for S100 proteins and other calcium sensor proteins [35,36,51]. Importantly, S100B monomer may be of physiological importance, considering that basal serum level of S100B is around 6 pM [52], which is close to the S100B homodimer dissociation constant [49].

### 2.2. Modulation of IFN-β Cytotoxicity towards MCF-7 Cells by S100B

Since the therapeutic use of IFN-β-1a raises its serum concentration up to 20 pM (100 IU/mL [53]), exceeding the serum S100B level of 6 pM [52], IFN-β binding to S100B (Table 1) could affect functioning of the latter. Similarly, the increase in serum S100B level under pathological conditions up to 0.2 nM [54], which exceeds the serum level of IFN-β, can promote its binding to S100B, possibly altering functional activities of IFN-β. Furthermore, S100B and IFN-β levels in CSF may reach 15 nM [55] and 2 pM [56], respectively.

To probe the ability of extracellular S100B to affect the cellular effects induced by IFN-β, we studied an influence of IFN-β and various S100B concentrations on viability of MCF-7 cells by crystal violet assay. In accordance with the previous reports [30,31,57,58,59], 200 pM IFN-β decreases viability of the cells by ca. 20% relative to the control (Figure 5B). S100B in concentrations from 5 nM to 25 nM (nearby the K_d_ estimated from the fluorometric data) also suppresses viability of MCF-7 cells by 14–28% (Figure 5A). Meanwhile, the combined action of S100B and IFN-β used in the same concentrations does not affect viability of MCF-7 cells (Figure 5B). These observations indicate that the formation of the complex between IFN-β and S100B compensates their cytotoxic effects. Therefore, S100B binding to IFN-β seems to reverse the activity of the latter against MCF-7 cells.

Furthermore, an increase in S100B concentration up to 100–200 nM is accompanied by the loss of its effect on the viability of MCF-7 cells (Figure 5A) and recovery of the inhibitory activity of IFN-β under combined use with S100B (Figure 5B). Hence, the elevated S100B level disrupts the non-additivity of the cellular effects induced by IFN-β and S100B, possibly due to the formation of some higher-order multimers of S100B [60]. Overall, IFN-β and 5–25 nM S100B exert non-additive effects on the viability of MCF-7 cells, thereby evidencing the S100B binding to IFN-β, which reverses the IFN-β activity towards the cells. The inhibition of the IFN-β-induced suppression of viability of MCF-7 cells was previously shown also for S100A1/A4/P proteins [30,31].

### 2.3. Human Diseases Associated with Dysregulation of IFN-β and S100B

The effects of S100B on IFN-β signaling (Figure 5) demonstrate their potential physiological significance. IFN-β is secreted in various diseases [1], some of which could be accompanied by the elevated S100B levels, thereby favoring its binding to IFN-β. We searched for the human diseases associated with involvement of IFN-β and S100B in DisGeNET and Open Targets Platform (‘OTP’) databases.

DisGeNET contains 88 diseases related to both S100B and IFN-β, including carcinomas of breast, lung, colon and stomach, melanoma, neuroblastoma, glioma, multiple sclerosis, Alzheimer’s disease, schizophrenia, depressive disorder, mental depression, impaired cognition, etc. (Appendix A).

The OTP database includes 225 entries associated with both S100B and IFN-β (Appendix A). Consideration of the cases with association scores above 0.1 reveals the following entries: neoplasm, cancer, brain and skin neoplasms, glioma, melanoma, inflammation, vascular disease, type II hypersensitivity reaction disease, injury, infectious disease, viral disease, lung disease, liver disease, stroke, neurodegenerative disease, nervous system disease, central nervous system disease, neuropathy, peripheral neuropathy, mental or behavioral disorder, and cognitive disorder.

Summing up, this bioinformatics analysis points out that the direct effect of S100B protein on IFN-β activity is of potential relevance to pathogenesis of multiple oncological and neurological diseases. For instance, the elevated levels of S100B in several cancers [47] may interfere with the antiproliferative effects of IFN-β with regard to cancer cells, as exemplified by MCF-7 cells (Figure 5).

### 2.4. Modeling of S100B–IFN-β Complexes

Modeling of the quaternary structure of the complex between the IFN-β and Ca^2+^-loaded S100B monomer using ClusPro docking server [61] predicts that the residues of IFN-β helix I (Arg11, Phe15, Gln18, Lys19, Trp22), loop between helices II and III (Gln48), helix VI (Arg147, Val148, Leu151, Arg152, Tyr155, Asn158, Arg159, Gly162) and C-terminus (Arg165) bind to the area formed by the residues of helix I (Met1, Glu3, Glu5, Lys6), helix IV (Met75, His86) and C-terminus (Glu92) of the S100B monomer (Figure 6A). The analogous modeling for the Ca^2+^-loaded S100B dimer predicts that its chain A interacts with IFN-β via helix I (Asp13), while chain B of the dimer binds IFN-β via the ‘hinge’ region between helices II and III (His43, Phe44), helix VI (Glu87, Phe88, Phe89) and C-terminus (Glu90) (Figure 6B). The respective binding area of IFN-β is formed by residues of helix I (Phe15, Trp22), loop between helices I and II (Arg27, Arg35), and helix VI (Arg147, Val148, Leu151, Arg152, Tyr155) (Figure 6B). Notably, the residues His43, Phe44, and Phe88 of the S100B chain B are also involved in the recognition of the RAGE peptide (PDB entry 5D7F).

The tighter contact with IFN-β observed in the model structure for the monomeric S100B (Figure 6) is in line with its 2.7 orders of magnitude higher affinity to IFN-β (Table 1), compared to that of the S100B dimer. The proximity of the Ca^2+^-binding loops of S100B to the contact surface in both model structures (Figure 6) could rationalize calcium sensitivity of the S100B–IFN-β interactions (Figure 3).

Inspection of the crystal structure of human IFN-β (PDB entry 1AU1) reveals that Trp79 and Trp143 are buried in the protein interior, whilst Trp22 is highly accessible to the solvent (Figure 6). Therefore, the long wavelength position of tryptophan fluorescence emission maximum of IFN-β, *λ_max_*, ca. 342 nm (Figure 3A), shows that Trp22 provides a major contribution to the fluorescence emission spectrum. Hence, the S100B-induced changes in fluorescence emission spectrum of IFN-β (Figure 3) indicate that the S100B–IFN-β interaction affects the local environment of the Trp22. The modeling of the S100B–IFN-β complex supports this conclusion (Figure 6).

### 2.5. Intrinsic Disorder Propensity of S100B

Earlier, based on the comprehensive bioinformatics analysis of 81 S100 proteins subdivided into 22 groups, we concluded that these proteins are enriched in predicted intrinsic disorder [63]. In fact, 46 S100 proteins (57%) from 14 groups were predicted to be highly disordered, with their average disorder score values evaluated by PONDR^®^ VSL2 (ADS_VSL2_) exceeding the threshold of 0.5. The ADS_VSL2_ values for all 81 sequences of S100 analyzed in that study ranged from 0.289 to 0.875, and the average ADS_VSL2_ for the entire set was 0.51 ± 0.12. The ADS_VSL2_ values for 46 highly disordered S100 family members ranged from 0.505 to 0.875 and their average ADS_VSL2_ was 0.59 ± 0.08 [63].

We also pointed out that the intrinsic disorder propensity is unevenly distributed within the S100 sequences, with the central region of ~50 residues being preferentially disordered. Curiously, this disordered region typically includes the major part of the pseudo EF-hand loop, helix II, hinge region, and an initial part of helix III, and also acts as a major target for the enzymatic post-translational modifications [63]. Furthermore, the C-terminal half of the central region including the hinge and the N-terminal half of helix III is used by the S100 proteins for target recognition, with intrinsic disorder of this segment being crucial for partner recognition. In fact, the S100 proteins predicted to be disordered were shown to be characterized by high binding promiscuity, being involved in interaction with at least 86% of known S100 binding partners [63]. Since S100B has an ADS_VSL2_ of 0.51 ± 0.17, it is one of the 46 highly disordered members of the S100 family and therefore is expected to be a promiscuous binder as well.

Recently, we also showed that the S100 proteins capable of IFN-β binding (S100A1/A4/A6/P) are characterized by a remarkable similarity of their disorder profiles, whereas the disorder profiles of S100B/A7/A8/A9/A10/A11/A12/A13/A14/A15, which did not bind IFN-β, were not only highly diversified within the group, but were generally very different from the profiles of the IFN-β binders [31]. The only exception from this rule was S100B, whose disorder profile was very similar to the profiles of the four IFN-β binders [31]. Figure 7A illustrates this observation by comparing the PONDR^®^ VSL2-generated disorder profile of S100B with those of S100A1, S100A4, S100A6, and S100P. This remarkable similarity of disorder propensities is in line with close evolutionary similarity of S100B to S100A1 and S100P [31]. This also strongly suggests that the binding promiscuity in general and the capability of interaction with IFN-β in particular are somehow encoded in the peculiarities of the distribution of intrinsic disorder predisposition within the amino acid sequences of S100 proteins.

Finally, looking at the model complexes between the S100B and IFN-β (Figure 6) from the intrinsic disorder angle provides further support to the functional importance of disorder, as the ‘hinge’ region between helices II and III, which is predicted to be disordered (Figure 7), plays a crucial role in the complex formation. Multiple sequence alignments of human S100B, S100A1, S100A4, S100A6, and S100P by Clustal Omega [64] revealed that these proteins share relatively low sequence identity, which ranges from 34.83% to 57.61% (see Appendix A). However, Figure 7B, which represents these proteins in the form of sequence logo, clearly indicates strong conservation of some positions within the disordered binding region (residues 20–70), suggesting potential role of these residues in the functionality of these promiscuous S100 proteins.

## 3. Materials and Methods

### 3.1. Materials

Human interferon beta-1a produced in CHO-K1 cells (IFN-β) was purchased from Merck (Rebif^®^). Human S100B was expressed in *E. coli* and purified as described in ref. [62]. Protein concentrations were measured spectrophotometrically according to ref. [64].

Sodium acetate, HEPES, NaOH, DTT and SDS were from PanReac AppliChem. Sodium chloride was from Helicon (Moscow, Russia). CaCl_2_, EDTA and TWEEN 20 were purchased from Sigma-Aldrich Co. NAP-5 column was from Cytiva.

ProteOn™ GLH sensor chip, amine coupling kit, EDAC and sulfo-NHS were from Bio-Rad Laboratories, Inc. (Hercules, CA, USA).

MCF-7 cell line was from European Collection of Authenticated Cell Cultures. Crystal violet was from Sigma-Aldrich Co.

### 3.2. Chemical Crosslinking

Crosslinking of IFN-β with S100B was performed mainly as described in ref [31]. A 2 mg/mL solution of S100B in 10 mM HEPES pH 8.0 buffer containing 20 mM DTT and 20 mM EDTA was passed through NAP-5 desalting column for calcium removal. The protein samples (5 µM IFN-β with S100B added up to IFN-β to S100B molar ratio of 1:0, 1:0.5, 1:1, 1:2, 0:2) were treated with 40 mM EDAC and 10 mM sulfo-NHS at 25 °C for 1.5 h (10 mM HEPES, 150 mM NaCl, pH 7.4, with 1 mM CaCl_2_ for Ca^2+^-bound S100B or without CaCl_2_ for Ca^2+^-depleted S100B). The reaction was quenched by addition of SDS-PAGE sample loading buffer, followed by SDS-PAGE (15%) with silver staining.

### 3.3. Fluorescence Measurements

Fluorescent titrations of IFN-β solution (0.5 µM) in 10 mM HEPES-NaOH, 150 mM NaCl, 1 mM CaCl_2_ or 5 mM EDTA, pH 7.4 buffer by S100B stock solution were performed at 25 °C mainly as previously described [30]. Since S100B lacks Trp residues, excitation at 295 nm (excitation monochromator bandwidth of 2.5 nm) ensures detection of fluorescence emission spectra of Trp residues of IFN-β. The spectra were corrected for spectral sensitivity of the fluorimeter and fitted by a log-normal function. The fluorescence spectrum maximum positions (*λ_max_*) were obtained from these fits. Fluorescence intensities were corrected for the dilution effect. Estimation of IFN-β affinity to S100B was performed using a one-site binding model as implemented in FluoTitr v.1.42 software (IBI RAS, Pushchino, Moscow region, Russia).

### 3.4. Surface Plasmon Resonance Studies

SPR measurements were performed at 25 °C using Bio-Rad ProteOn™ XPR36 instrument generally as described in ref. [31]. Ligand (50 μg/mL S100B in 10 mM sodium acetate pH 4.5 buffer) was immobilized on ProteOn™ GLH sensor chip surface (up to 9000 resonance units, RUs) by amine coupling. Analyte (10–80 nM IFN-β) in a running buffer (10 mM HEPES, 150 mM NaCl, 0.05% TWEEN 20, 1 mM CaCl_2_, pH 7.4) was passed over the chip, followed by flushing the chip with the running buffer. The sensor chip surface was regenerated by passage of 5 mM EDTA pH 8.0 solution for 200 s. The double-referenced SPR sensograms were globally fitted according to a heterogeneous ligand model, which assumes existence of two populations of a ligand (L_1_ and L_2_) that bind single analyte molecule (A):
K_d1_     K_d2_
L_1_ + A ↔ L_1_A; L_2_ + A ↔ L_2_A
k_d1_     k_d2_(1)
where K_d_ and k_d_ refer to equilibrium and kinetic dissociation constants, respectively. K_d_, k_d_ and R_max_ (maximum response) values were evaluated using Bio-Rad ProteOn Manager™ v.3.1 software (Hercules, CA, USA). The data are presented as the means ± standard deviations.

### 3.5. Cell Viability Studies

Viability of MCF-7 cells treated by S100B (0–200 nM), IFN-β (200 pM) or their combination was measured by crystal violet assay as previously described [31]. The data are presented as the means ± standard deviations (*n* = 5–6).

### 3.6. Search of Diseases Associated with IFN-β and S100B

The data on diseases associated with both genes IFNB1 (human IFN-β, UniProt ID P01574) and S100B (human S100B, UniProt ID P04271) were collected from the human disease databases DisGeNET v.7.0 [67] and Open Targets Platform v.20.09 [68] as described in ref. [30]. The DisGeNET entries were manually curated; false positive records were removed.

### 3.7. Modeling of S100B–IFN-β Complexes

The models of tertiary structures of the S100B–IFN-β complexes were built using ClusPro docking server (https://cluspro.bu.edu/login.php accessed on 26 November 2021) [61], based on the crystal structures of human IFN-β and Ca^2+^-loaded human S100B dimer complexed with RAGE peptide extracted from PDB [69] entries 1AU1 (chain A) and 5D7F (chains A and B), respectively. The structure of Ca^2+^-loaded S100B monomer was predicted from the amino acid sequence using I-TASSER server (https://zhanggroup.org//I-TASSER/ accessed on 26 November 2021) [62]. Distributions of the contact residues in the docking models over the protein sequences were calculated using Python 3.3 programming language (https://www.python.org/ accessed on 26 November 2021) (implemented in PyCharm v.3.0.2 development environment), Matplotlib Python plotting library and NumPy numerical mathematics extension. The residues included into 6 or more docking models were considered as the most probable residues of the binding site. The models most closely covering the residues of the probable binding sites were taken. The tertiary structure models were drawn with molecular visualization system PyMOL v.1.6.9.0 (https://pymol.org/2/ accessed on 26 November 2021). The numbering of the contact residues is according to the PDB entries.

## 4. Conclusions

The in vitro study presented here shows that the S100 proteins, which specifically bind IFN-β, are not restricted to S100A1/A4/A6/P [30,31], but also include the S100B protein. Importantly, S100B is evolutionary fairly distant from S100A1/A4/A6/P proteins: the pairwise sequence identities between them calculated by Clustal2.1 range from 57.61% (S100B–S100A1 pair) to 39.33% (S100B–S100A6) (see Appendix A). Hence, the IFN-β–S100B interaction found in our study is non-redundant. Examination of amino acid sequence of IFN-β did not reveal the known consensus S100B binding motif found in intracellular and extracellular proteins of vertebrates [35].

The binding to S100B is the most strong interaction reported for IFN-β to date. The S100–IFN-β interaction requires Ca^2+^ excess and dramatically improves upon S100 monomerization. Notably, the IFN-β-specific S100 proteins belong to the category of ‘promiscuous’ S100 proteins, which are able to bind several ligands with a high affinity [70]. The other S100 proteins lacking specificity to IFN-β are considered as ‘orphan’ binders, which weakly bind to no more than one partner [70]. Thus, the revealed regularities of cross-interactions of the S100 proteins with IFN-β could be an inherent property of the S100 protein family, related to peculiarities of the distributions of intrinsic disorder predispositions within their amino acid sequences.

Noteworthy is that S100B is a much more effective inhibitor of the IFN-β activity towards MCF-7 cells (Figure 5), compared to S100A1/A4/P proteins [30,31]. In contrast to the inhibition of the IFN-β-induced suppression of MCF-7 cells viability by S100A1/A4/P, IFN-β complexed with S100B favors viability of MCF-7 cells. Nevertheless, the antagonistic effects of the INF-β-specific S100 proteins indicate amplification of their concerted action. Therefore, a directed inhibition of the S100 proteins is expected to increase therapeutic efficacy of IFN-β. This approach could be advantageous for the oncological diseases, accompanied by increased S100A1/A4/A6/P/B expression [1,47,71]. The presented bioinformatic analysis of the human diseases associated with simultaneous involvement of IFN-β and S100B shows that the IFN-β–S100B interaction is also relevant to pathogenesis of numerous neurological diseases.

## Figures and Tables

**Figure 1 ijms-23-01997-f001:**
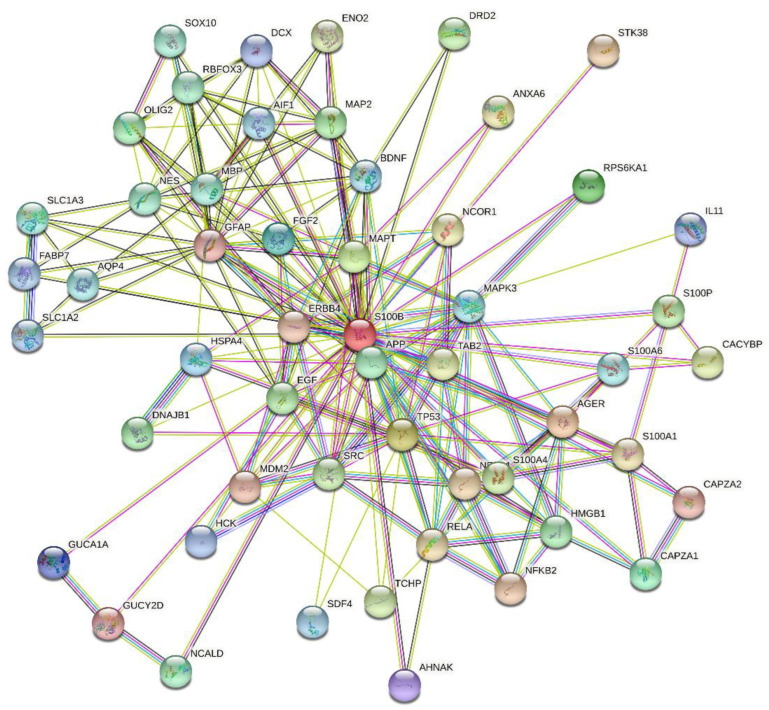
S100B-centered protein–protein interaction (PPI) network generated by STRING using high confidence level of 0.7 [47]. In this network, the nodes correspond to proteins, whereas the edges show predicted or known functional associations. Seven types of evidence are used to build the corresponding network, and are indicated by the differently colored lines: a green line represents neighborhood evidence; a red line—the presence of fusion evidence; a purple line—experimental evidence; a blue line—co-occurrence evidence; a light blue line—database evidence; a yellow line—text mining evidence; and a black line—co-expression evidence [47]. This network includes 53 proteins connected by 205 interactions. As a result, an average node degree of this network is 7.74.

**Figure 2 ijms-23-01997-f002:**
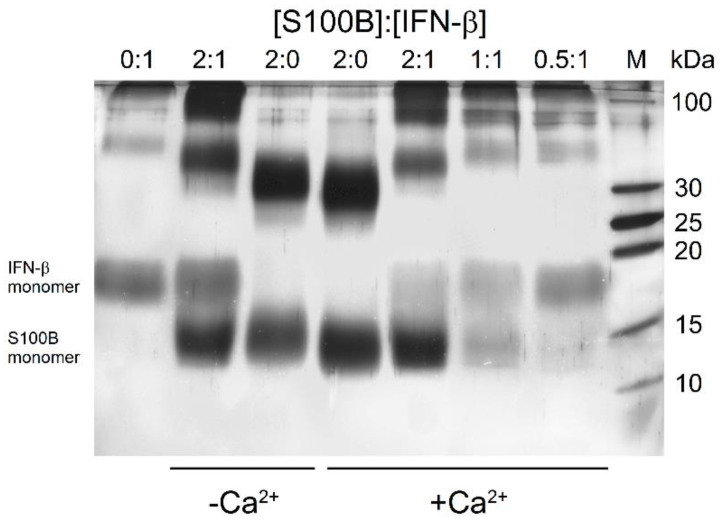
SDS-PAGE results for 5 μM IFN-β with/without Ca^2+^-depleted/bound S100B (S100B to IFN-β molar ratios are indicated) cross-linked with EDAC/sulfo-NHS at 25 °C.

**Figure 3 ijms-23-01997-f003:**
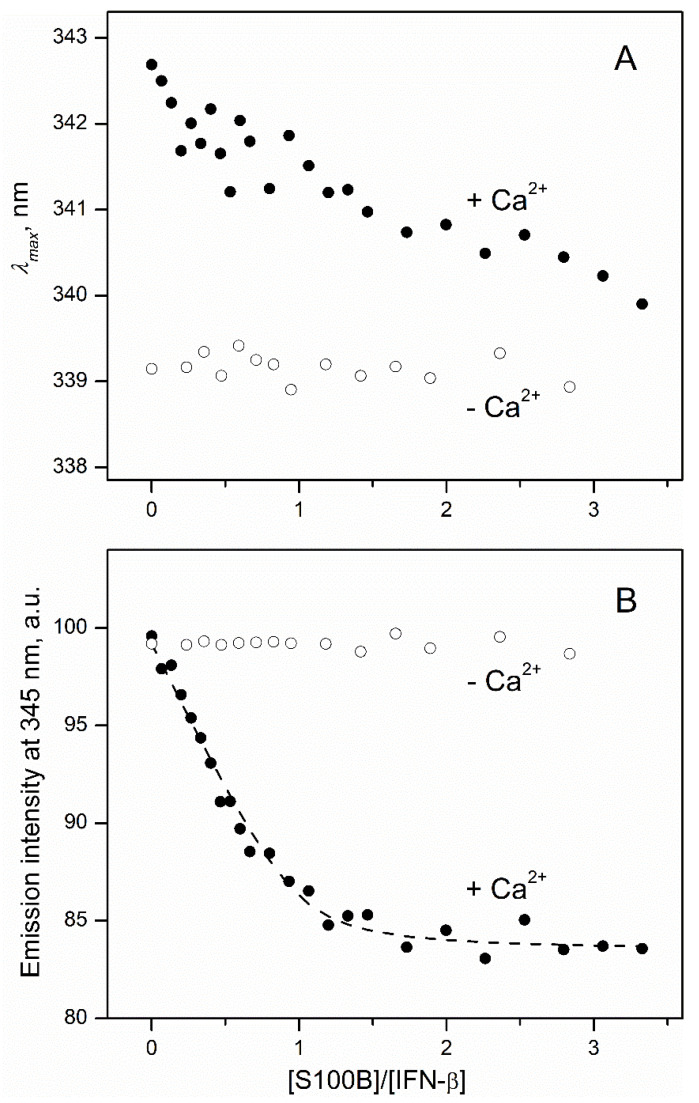
IFN-β interaction with S100B at 25 °C (1 mM CaCl_2_ or 5 mM EDTA, pH 7.4), monitored by tryptophan fluorescence of IFN-β (0.5 μM; excitation at 295 nm). (**A**) Spectrum maximum position; (**B**) emission intensity at 345 nm. The points are experimental, while the dashed curve is theoretical, fitted according to the one-site binding model.

**Figure 4 ijms-23-01997-f004:**
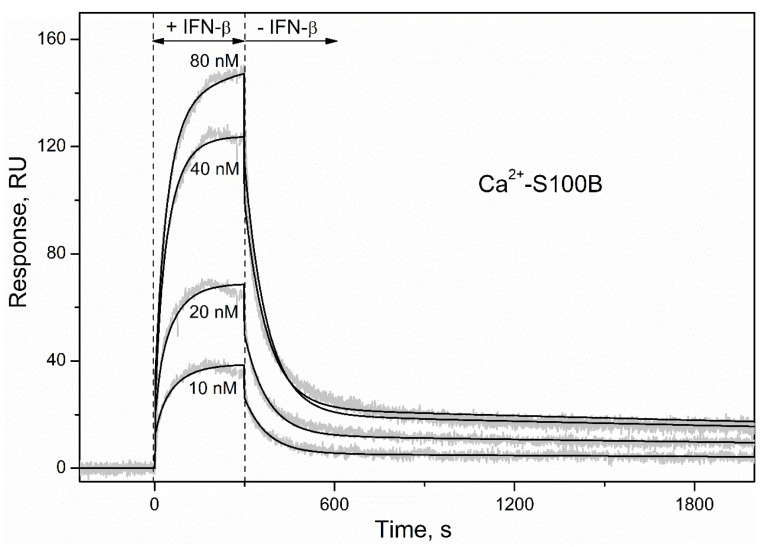
Kinetics of the interaction between IFN-β and Ca^2+^-bound S100B at 25 °C (1 mM CaCl_2_, pH 7.4), monitored by SPR spectroscopy using IFN-β as an analyte (10–80 nM) and S100B as a ligand. Grey curves are experimental, while black curves are theoretical, calculated according to the heterogeneous ligand model [1] (see Table 1 for the fitting parameters).

**Figure 5 ijms-23-01997-f005:**
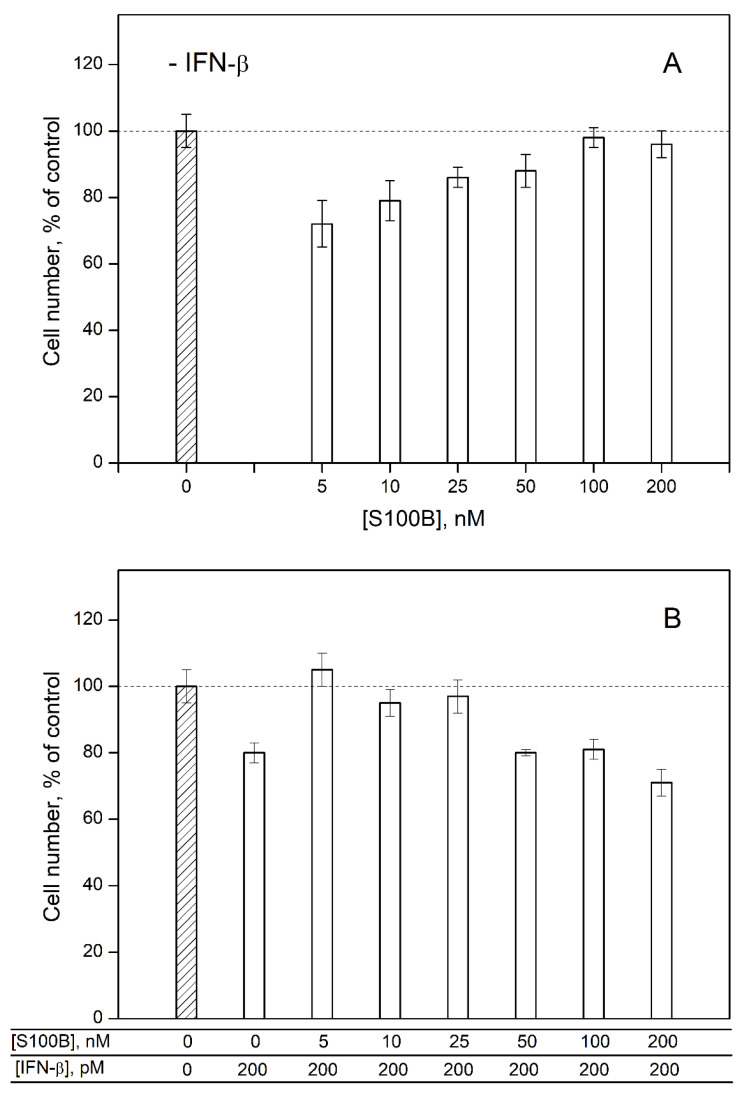
The data of crystal violet assay for viability of MCF-7 cells treated by S100B (panel (**A**)), IFN-β or their combination (**B**).

**Figure 6 ijms-23-01997-f006:**
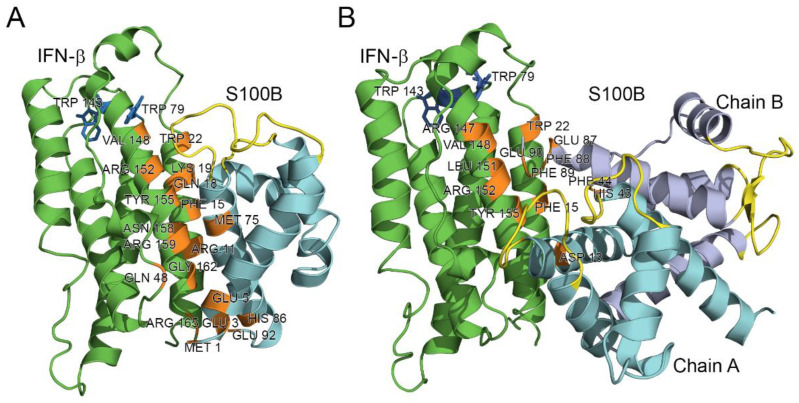
The models of tertiary structures of IFN-β (shown in green) bound to Ca^2+^-loaded S100B monomer (**A**) or dimer (**B**). Crystal structures of IFN-β and Ca^2+^-loaded S100B dimer (PDB entries 1AU1 and 5D7F, respectively) are used for the modeling by ClusPro docking server [61]. The model of S100B monomer is built using I-TASSER server [62]. The Ca^2+^-binding loops are shown in yellow. The numbering of the contact residues (orange-colored) is according to the PDB entries.

**Figure 7 ijms-23-01997-f007:**
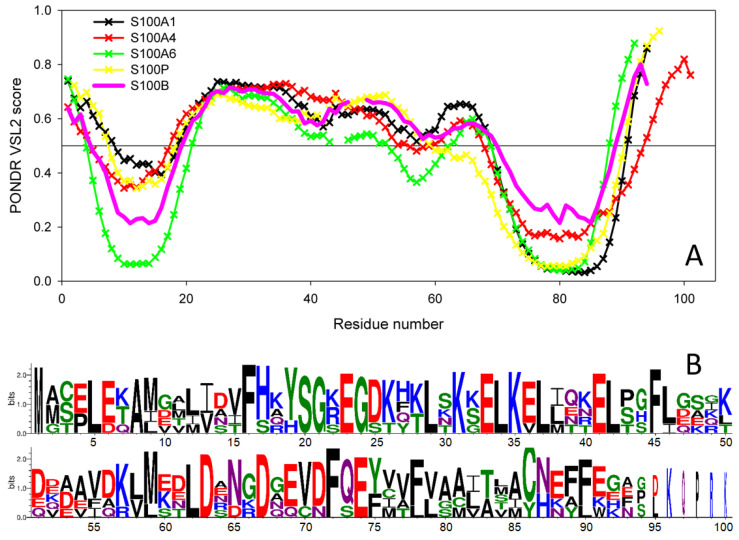
(**A**) Intrinsic disorder predisposition of human S100B protein in comparison with the disorder profiles of human S100 proteins capable of IFN-β binding, estimated using PONDR^®^ VSL2 algorithm (http://www.pondr.com/ accessed on 26 November 2021). The plot represents data for the sequences aligned using Clustal Omega [64], and breaks in the curves correspond to the alignment gaps. (**B**) Sequence logo representation of human S100B, S100A1, S100A4, S100A6, and S100P proteins generated by the WebLogo tool (http://weblogo.threeplusone.com/ accessed on 26 November 2021) [65,66].

**Table 1 ijms-23-01997-t001:** Parameters of the heterogeneous ligand model [1] describing the SPR data on kinetics of the IFN-β association with Ca^2+^-loaded S100B (Figure 4).

k_d1_, s^−1^	K_d1_, pM	R_max1_	k_d2_, s^−1^	K_d2_, nM	R_max2_
(1.49 ± 0.12) × 10^−4^	47 ± 34	20	(1.3 ± 0.6) × 10^−2^	87 ± 24	217

## Data Availability

The data presented in this study are available within the article and Appendix A.

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
