# Peer review of "Interferon-β Activity Is Affected by S100B Protein"

_ijms, 2022, doi:10.3390/ijms23041997_

Round 1

Reviewer 1 Report

The revised manuscript presented by the authors is very similar to the previous articles, published by the same authors, in which the interaction of IFN-β with S100 proteins is described. It is therefore not surprising that the S100B protein, belonging to this family, also interacts with IFN-β. In my opinion this work has little added value.

Authors should try to introduce innovative aspects into the research in order to differentiate this manuscript from previous ones. So, I recommend the manuscript to be accepted for publication after major revision.

Author Response

The revised manuscript presented by the authors is very similar to the previous articles, published by the same authors, in which the interaction of IFN-β with S100 proteins is described. It is therefore not surprising that the S100B protein, belonging to this family, also interacts with IFN-β. In my opinion this work has little added value.

ANSWER: We have emphasized in the first of paragraph of the Conclusions section that pairwise sequence identities between S100B and other S100 proteins shown to interact with IFN-β do not exceed 57.61%. Thus, they substantially differ from each other, and therefore not necessarily interact with the same target proteins. The IFN-β – S100B is the strongest interaction reported for IFN-β to date, accompanied by the most vivid effects in the cellular assay with MCF-7 cells. For this reason, we disagree that this interaction is less significant than the previously reported interactions.

Authors should try to introduce innovative aspects into the research in order to differentiate this manuscript from previous ones. So, I recommend the manuscript to be accepted for publication after major revision.

ANSWER: We have changed the Conclusions chapter to emphasize uniqueness of the revealed interaction in comparison with other IFN-β – S100 interactions reported to date. We also modified Figure 7 to include additional information on the peculiarities of S100B sequence.

Reviewer 2 Report

The manuscript of Kazakov et al. “Interferon-β activity is affected by S100B protein” is devoted to a multidisciplinary approach in order to evaluate a protein from the S100 family relevant to several oncological and neurological diseases.

In general, the manuscript is well organized and documented, and presented with enough clarity.

This reviewer would like to recommend the following corrections before publication:

Abstract, lines 19 and 23: Ca2+, 2+  should be corrected to superscript.

Introduction: The sentence starting on line 46 should be revised.

Figure 1 legend: this reviewer suggest the following modification in line 111: SDS-PAGE results instead of “the results of SDS-PAGE”.

Figure 2 legend: this reviewer suggest the following modification in line 159: INF-β interaction instead of “the interaction of INF-β”.

Modeling section: the authors should use the term “quaternary” and “tertiary” structure more carefully when taking into account their modelled structures and complexes.

Figure 5: INF-b should be corrected to INF-β.

Authors claim that Trp79 and Trp143 are buried, but they are not highlighted in the Figure 5. This reviewer suggest to introduce them in the Figure, so, the general audience can see the point easily.

Line 256: PDB entry 1au1, au should be corrected to capital letters.

Intrinsic disorder propensity of S100B section. This reviewer suggest to introduce in the text of the manuscript the main nature composition of the ca. 50 disordered residues relevant for recognition.

References 62-131 require revision. Actually they started with numbering 1 at reference position 63, which actually constitutes a repetition from the initial ones.

Author Response

The manuscript of Kazakov et al. “Interferon-β activity is affected by S100B protein” is devoted to a multidisciplinary approach in order to evaluate a protein from the S100 family relevant to several oncological and neurological diseases. 

In general, the manuscript is well organized and documented, and presented with enough clarity. 

This reviewer would like to recommend the following corrections before publication: 

Abstract, lines 19 and 23: Ca2+, 2+  should be corrected to superscript. 

ANSWER: Corrected

Introduction: The sentence starting on line 46 should be revised. 

ANSWER: We have rephrased this sentence.

Figure 1 legend: this reviewer suggest the following modification in line 111: SDS-PAGE results instead of “the results of SDS-PAGE”.

ANSWER: Done

Figure 2 legend: this reviewer suggest the following modification in line 159: INF-β interaction instead of “the interaction of INF-β”.

ANSWER: Done

Modeling section: the authors should use the term “quaternary” and “tertiary” structure more carefully when taking into account their modelled structures and complexes.

ANSWER: We have replaced “quaternary” by “tertiary”.

Figure 5: INF-b should be corrected to INF-β.

ANSWER: We have corrected the label.

Authors claim that Trp79 and Trp143 are buried, but they are not highlighted in the Figure 5. This reviewer suggest to introduce them in the Figure, so, the general audience can see the point easily.

ANSWER: We have added Trp79 and Trp143 to Figure 6 and inserted the respective reference into the manuscript text.

Line 256: PDB entry 1au1, au should be corrected to capital letters.

ANSWER: Done

Intrinsic disorder propensity of S100B section. This reviewer suggest to introduce in the text of the manuscript the main nature composition of the ca. 50 disordered residues relevant for recognition. 

ANSWER: Thank you for pointing this out. We included a new plot to Figure 7 showing peculiarities of the S100B sequence in comparison with four other promiscuous S100 proteins, S100P, S100A1, S100A4, and S100A6 in a form of sequence logo. Corresponding discussion is also added to the revised manuscript.

References 62-131 require revision. Actually they started with numbering 1 at reference position 63, which actually constitutes a repetition from the initial ones.

ANSWER: Corrected

Reviewer 3 Report

In the manuscript of Alexey S. Kazakov at al. the influence of S100B protein on the activity of interferon-β (INF-β) and interaction between these proteins have been studied. Interferon-β is a pleiotropic cytokine used for therapy of multiple sclerosis, it also affects the progressing of cancer and some other diseases. S100B protein, involving in the progress of numerous oncological and neurological diseases, belongs to S100 family of multifunctional Ca2+-binding proteins possessing cytokine-like activities. Some of S100 proteins are able to interact with interferon-β. INF-β and S100 proteins are involved in the regulation of many processes in the cell, so it is important to understand the mechanism of their interaction. As it is shown in the manuscript the interaction of IFN-β with S100B is the most strong reported for IFN-β to date. The interaction between S100B and IFN-β and its influence on the regulatory activity of these proteins were characterized by several methods. It was shown that binding of these proteins requires Ca2+ excess and depends on the oligomeric (monomeric or dimeric) state of S100B protein. The modeling of the structure of complex IFN-β with Ca2+-loaded S100B monomer and dimer allowed to characterize the contact area between the complex components. Using the bioinformatic analysis of the human diseases associated with involvement of IFN-β and S100B it was shown that IFN-β – S100B interaction is relevant to numerous neurological diseases. The multiple sequence alignments of some S100 proteins revealed that disordered fragments of the srtucture which are typical for these proteins contain invariant amino acid residues playing a crucial role in the interaction with other molecules upon complex formation. The manuscript of A. S. Kazakov et al. contains an essential new information on the interaction between two cytokines and can be published with minor correction.

Remarks

line 171-172, page 5. It is written:IFN-β binds S100B depending on the S100B conformation, with a preference to the monomeric Ca2+-bound form”. It is not clear what means “S100B conformation”. May be “depending on the S100B oligomeric state” ?

Page 3. In the section entitled “Conformation-dependent interaction between IFN-β and S100B” the interaction between IFN-β and S100B is considered depending on the oligomeric state (dimer or monomer) of the S100B molecule . This title suggests, that the conformational changes in the molecule of S100B accompanying the complex formation will be described. However, with regard to conformation, it is only written near the end of this section that “The strong calcium dependence of this interaction is likely due to the Ca2+-induced solvent exposure of hydrophobic surfaces of S100B, typical for S100 proteins and other calcium sensor proteins [35, 36, 51]”. So the conformation of the molecule is not discussed here. Page 8. Modeling of S100B—IFN-β complexes. Line 238 ” Modeling of the tertiary structure of the complex More correct “quaternary structure of the complex” because the contact area between the conplex components is discussed mainly.

Author Response

line 171-172, page 5. It is written: “IFN-β binds S100B depending on the S100B conformation, with a preference to the monomeric Ca2+-bound form”. It is not clear what means “S100B conformation”. May be “depending on the S100B oligomeric state” ?

ANSWER: We have rephrased the sentence: “Overall, IFN-β binds S100B depending on Ca2+ and oligomeric state of S100B, with a preference to the monomeric Ca2+-bound form”.

Page 3. In the section entitled “Conformation-dependent interaction between IFN-β and S100B” the interaction between IFN-β and S100B is considered depending on the oligomeric state (dimer or monomer) of the S100B molecule . This title suggests, that the conformational changes in the molecule of S100B accompanying the complex formation will be described. However, with regard to conformation, it is only written near the end of this section that “The strong calcium dependence of this interaction is likely due to the Ca2+-induced solvent exposure of hydrophobic surfaces of S100B, typical for S100 proteins and other calcium sensor proteins [35, 36, 51]”. So the conformation of the molecule is not discussed here.

ANSWER: We have changed the title of this section: “High-affinity interaction between IFN-β and S100B”.

Page 8. Modeling of S100B—IFN-β complexes. Line 238 ” Modeling of the tertiary structure of the complex“ More correct “quaternary structure of the complex” because the contact area between the complex components is discussed mainly.

ANSWER: Corrected

Reviewer 4 Report

The authors showed for the first time that IFN-β interacts with the S100B protein, and using advanced analytical techniques, they approximated the mechanism of these interactions. These findings could play a role in new approaches to the treatment of numerous oncological and neurological diseases. The article is carefully prepared and meets the requirements of the IJMS editorial staff and should be published in the International Journal of Molecular Sciences after adjusting it to the reader's expectations and removing only a few minor correction errors, listed below:

1) The authors in a concise publication use a lot of specialized markings and abbreviations well known to authors and a narrow group of specialists. In order to facilitate the study reader, please add a new chapter, Abbreviations.

Comment: The reader of this wide-ranging journal IJMS needs to summarize / decode abbreviations independently in a separate chapter. List of the examples of abbreviations to be included in the new chapter: IFN-β, S100B, S100A1, S100A4, S100A6, MCF-7, DisGeNET, MCF-7, IFNAR1-IFNAR2, IF-NAR1, JAK-STAT, PPI, sulfo-NHS, SDS-PAGE, SPR, S100A1/A4/P, OTP, RAGE, PDB, PTMs, ADSVSL2, PONDR® VLS2, CHO-K1, HEPES, DTT, SDS, TWEEN 20, NAP-5, ProteOn™ GLH, HEPES, EDAC, IFNB1, etc.

2) At line 394 is § 5, but is lack of the § 4.

3) Please revise and standardize the literature cited, for example:

Ref. 7. Article number or pages please add.

Ref. 11. Article number or pages please add.

Ref 18. Article number or pages please add.

Ref 22. Article number or pages please add.

Ref 24. Article number please add.

Ref. 26. Please add end page.

Ref 27. Article number or pages please add.

Ref 29. At … Advances in clinical chemistry … , please use first capital letter style.

Ref 30. At … International journal of biological macromolecules … please use first capital letter style.

Ref. 31. Article number please add.

Ref. 18 and 31. At: … International journal of molecular sciences … , please use first capital letter style.

Ref. 36. Article number or pages please add.

Author Response

1) The authors in a concise publication use a lot of specialized markings and abbreviations well known to authors and a narrow group of specialists. In order to facilitate the study reader, please add a new chapter, Abbreviations.

Comment: The reader of this wide-ranging journal IJMS needs to summarize / decode abbreviations independently in a separate chapter. List of the examples of abbreviations to be included in the new chapter: IFN-β, S100B, S100A1, S100A4, S100A6, MCF-7, DisGeNET, MCF-7, IFNAR1-IFNAR2, IF-NAR1, JAK-STAT, PPI, sulfo-NHS, SDS-PAGE, SPR, S100A1/A4/P, OTP, RAGE, PDB, PTMs, ADSVSL2, PONDR® VLS2, CHO-K1, HEPES, DTT, SDS, TWEEN 20, NAP-5, ProteOn™ GLH, HEPES, EDAC, IFNB1, etc.

ANSWER: We have added the Abbreviations chapter into the manuscript text

2) At line 394 is § 5, but is lack of the § 4.

ANSWER: Corrected

3) Please revise and standardize the literature cited, for example:

Ref. 7. Article number or pages please add.

Ref. 11. Article number or pages please add.

Ref 18. Article number or pages please add.

Ref 22. Article number or pages please add.

Ref 24. Article number please add.

Ref. 26. Please add end page.

Ref 27. Article number or pages please add.

Ref 29. At … Advances in clinical chemistry … , please use first capital letter style.

Ref 30. At … International journal of biological macromolecules … please use first capital letter style.

Ref. 31. Article number please add.

Ref. 18 and 31. At: … International journal of molecular sciences … , please use first capital letter style.

Ref. 36. Article number or pages please add.

ANSWER: The references have been thoroughly checked and corrected

This manuscript is a resubmission of an earlier submission. The following is a list of the peer review reports and author responses from that submission.

Round 1

Reviewer 1 Report

The manuscript “Interferon-β Activity Is Affected by S100B Protein” by Kazakov et al. should be an interesting study on the interaction between IFN-β and the S100B protein, however the comparison with the previous work published in this same journal requires the authors to completely review the structure of this article to present an interesting scientific work to the wider public. So, I recommend the manuscript to be accepted for publication after revision.

Some other issues:

  • The authors should expand the introduction on the importance and properties of the S100B protein.
  • In figure 1 the authors should add the protein calibration standard used. Furthermore, the low resolution in the range 30-100 kDa prevents a correct analysis of the dimers of the S100B protein and of IFN-β cross-linked proteins. An analysis performed using a low-density gel should be added, as well as the authors should characterize by MS the cross-linked products.
  • Fluorescence studies completely lack the analysis of the effects of the Ca ions on the fluorescence emission of IFN-β. The authors should analysis and discuss why in absence of Ca ions the maximum fluorescence of IFN-β is at 339 nm, 4 nm lower than that recorded in the presence of Ca (Figure 2A, concentration of S100B = 0), greater than the blue shift recorded following the addition of the S100B protein (3 nm). The structure of IFN-β is influenced by the presence of Ca ions, the authors should add conformational data (Circular Dichroism, NMR, or other) on the IFN-β in presence or not of Ca ions.
  • The authors also do not carry out any studies to identify if all the three Trp residues present in the IFN-β protein are involved in the quenching process. An analysis using the Stern-Volmer equation can provide useful information in this regard.
  • I suggest to the Authors not to indicate prominent a blue shift of 3 nm (line 99)

Reviewer 2 Report

The work presented for review is very similar to the two previous papers (Kozakov et al., 2020 doi: 10.1016/j.ijbiomac.2019.12.039; Kozakov et al., 2020 doi: 10.3390/ijms21249473) in which the interaction of IFN-β with S100A1, S100A4, S100A6, and S100P is described in detail. All of the indicated proteins belong to the S100 protein family, so it is not surprising that another protein belonging to this family, namely S100B, also interacts with IFN-β. In my opinion, the work has little added value.

In addition, I also have other doubts that are indicated below.

Why are the other concentrations used in Figure 1 for studies with and without calcium ions? Changing the concentrations makes it difficult to compare the two results because the observed effect is not necessarily influenced by ions, but by changing the concentration ratio.

It is worth considering here the use of antibodies that would unambiguously show the protein shift. Describing these data in the form proposed in the manuscript is based only on the authors' guesses.

The authors mention that the differences in KD constants may be due to the monomeric state of the protein. Has any attempt been made to analyze the obtained sensorgrams using the 1: 1 model? This seems to be justified. Using a model delineating the two binding sites may result in inappropriate constants. Moreover, perhaps the immobilization of the protein on the chip causes one of the binding sites to be blocked. Has this hypothesis been tested?

SPR analysis and fluorimetric titration were carried out at 25°C. Why were such conditions chosen? Temperature influences the binding kinetics, and in this case, it seems more reasonable to conduct analyzes at 37°C, which is a temperature close to the temperature inside the host organism.

In the case of SPR analyzes, measurements were made in a buffer containing Ca ions. Have such analyzes been performed without calcium ions? Has a similar effect to fluorimetric titration been observed?

There is no information on the reference targets used in the part of the SPR methodology.